# Bimetallic TiO_2_ Nanoparticles for Lignin-Based Model Compounds Valorization by Integrating an Optocatalytic Flow-Microreactor

**DOI:** 10.3390/molecules27248731

**Published:** 2022-12-09

**Authors:** Swaraj Rashmi Pradhan, Marta Paszkiewicz-Gawron, Dariusz Łomot, Dmytro Lisovytskiy, Juan Carlos Colmenares

**Affiliations:** Institute of Physical Chemistry, Polish Academy of Sciences, Kasprzaka 44/52, 01-224 Warsaw, Poland

**Keywords:** bimetallic titanium dioxide, photocatalytic selective oxidation, fluoropolymer, wall-coated microreactor, aromatic alcohol, ultrasonic irradiation

## Abstract

The challenge of improving the activity of TiO_2_ by modifying it with metals and using it for targeted applications in microreactor environments is an active area of research. Recently, microreactors have emerged as successful candidates for many photocatalytic reactions, especially for the selective oxidation process. The current work introduces ultrasound-assisted catalyst deposition on the inner walls of a perfluoro-alkoxy alkane (PFA) microtube under mild conditions. We report Cu-Au/TiO_2_ and Fe-Au/TiO_2_ nanoparticles synthesized using the sol–gel method. The obtained photocatalysts were thoroughly characterized by UV–Vis diffuse-reflectance spectroscopy (DRS), high-resolution scanning electron microscopy (HR-SEM), high-resolution transmission electron microscopy (HR-TEM), X-ray diffraction analysis (XRD), Fourier transform infrared spectroscopy (FTIR), X-ray photoelectron spectroscopy (XPS), and N_2_ physisorption. The photocatalytic activity under UV (375 nm) and visible light (515 nm) was estimated by the oxidation of lignin-based model aromatic alcohols in batch and fluoropolymer-based flow systems. The bimetallic catalyst exhibited improved photocatalytic selective oxidation. Herein, four aromatic alcohols were individually investigated and compared. In our experiments, the alcohols containing hydroxy and methoxy groups (coniferyl and vanillin alcohol) showed high conversion (93% and 52%, respectively) with 8% and 17% selectivity towards their respective aldehydes, with the formation of other side products. The results offer an insight into ligand-to-metal charge transfer (LMCT) complex formation, which was found to be the main reason for the activity of synthesized catalysts under visible light.

## 1. Introduction

TiO_2_ is used as a photocatalyst for the degradation of impurities, a component of self-cleaning coatings, cosmetics, and for the production of hydrogen in water decomposition reactions [1,2]. Despite the numerous application possibilities, however, its industrial use is very limited. TiO_2_ due to wide band gap is capable of absorbing only UV radiation, which constitutes only 3 to 5% of solar radiation. The consequence of this is the fact that TiO_2_ still does not have satisfactory photocatalytic efficiency under visible light irradiation [3,4]. The use of UV lamps as a radiation source, due to the high energy consumption, significantly increases the cost of the process, which is an important factor limiting the broader use of this method in removing pollutants on an industrial scale. Consequently, many studies focused on the visible response of TiO_2_ photocatalysts by: (i) doping with metal or non-metal ions, (ii) hetero-junctions creation with other semiconductors [5,6], (iii) sensitization [7], (iv) the formation of a surface complex with energy transfer [8,9] or (v) surface modification with noble metals nanoparticles [10]. By doping TiO_2_ with transition-metal ions (Fe, Cu, Co, Mn, and Ni) was reported to be effective for the enhancement of the photocatalytic activity by changing the banding structure of TiO_2_ due to the interstitial doping Ti^3+^ states as well as the formation of surface oxygen vacancies, which, together, cause the red-shift absorption edge of TiO_2_ [11,12,13,14]. Among the noble metals, gold pays more attention in selective photo-oxidation reactions under visible light irradiation [15]. In the past few decades, the rapid expansion of gold (Au) catalysis has developed many new approaches to the aerobic oxidation of alcohols [16,17]. However, bimetallic catalysts have been observed to outperform their monometallic counterparts in conventional heterogeneous catalysis [18,19]. The noble bimetallic nanoparticles have been extensively studied for application in photocatalysis [20].

Because of their high-value products for the fine chemical, agrochemical, and pharmaceutical sectors, alpha beta-unsaturated aldehydes are essential to a sustainable chemical economy [21]. For example, cinnamyl aldehyde (CinAld) serves as an insect repellent and also provides the flavor and aroma of cinnamon; CinAld is both a food and perfume additive. These aldehydes are usually derived by selective oxidation of their corresponding alcohols.

Catalyst testing in flow microreactors has many advantages over traditional batch reactors. These are, but not limited to, the followings: (i) operating parameters such as temperature, pressure, and feed concentrations can be easily varied in flow microreactors to obtain an insight into the reaction mechanism and kinetics [22]; (ii) the consumption of chemicals and waste production are significantly reduced; and (iii) easy testing of catalyst stability under different reaction conditions. Liquid phase catalytic oxidation chemistry in continuous-flow microreactors has recently been summarized from technological and chemical perspectives [23].

Considering the potential of metals containing TiO_2_ catalytic materials and flow microreactors as a powerful tool for catalyst testing, studying the bimetallic catalysts in a flow microreactor for photocatalytic selective oxidation of a lignin-based model compound into a value-added product is an exciting field of research. In this work, the photocatalytic activity of synthesized bimetallic TiO_2_ nanoparticles was evaluated under UV light and visible light irradiations both in batch and microflow systems, using benzyl alcohol (BnOH), vanillin alcohol (VanOH), cinnamyl alcohol (CinOH), and coniferyl alcohol (ConOH) as model compounds of organic-based waste, lignin. The structural properties of the synthesized catalysts were analyzed by X-ray diffraction (XRD) and Fourier transform infrared (FTIR) spectroscopy. The optical properties were investigated using UV–vis diffuse reflectance (DRS), a surface morphology study was performed by high-resolution scanning electron microscopy (HR-SEM) and high-resolution transmission electron microscopy (HR-TEM), and the textural properties were determined by nitrogen sorption. We demonstrated that the addition of a second metal-to-metal TiO_2_ nanoparticle increased the selective oxidation of BnOH to 100% towards benzaldehyde (BnAld). In addition, the complex formation between TiO_2_ and other aromatic alcohols (containing methoxy and hydroxyl groups) activated the sol–gel-synthesized catalyst under visible light. Under batch experiment conditions, ConOH and VanOH showed conversion of 93% and 52%, respectively, under visible light (515 nm).

## 2. Results and Discussion

Oxidation of benzyl alcohol (BnOH) was often used as a model reaction for the oxidation of aromatic alcohols. In order to verify the reactivity of the prepared Cu-Au/TiO_2_ and Fe-Au/TiO_2_ catalysts in the batch system and the performance in the microreactor system, oxidation of BnOH was initially carried out under UV light (375 nm). From the photocatalytic experiment, it was found that modification of TiO_2_ with bimetals increased the selective oxidation of BnOH to aldehydes under UV light. In a batch system, the synthesized Cu-Au/TiO_2_ with copper acetate precursor (CuA-Au/TiO_2_) catalyst has a significantly higher conversion of BnOH (60%) compared to other bimetallic catalysts with 100% selectivity towards benzaldehyde (BnAld) (Figure 1a).

Upon the deposition of catalysts on the walls of the microreactor, the synthesized bimetallic materials revealed better photo reactivity in regard to both BnOH specific conversion rates, an absolute contrast trend with the batch experiments (Figure 1a). After 3 h of light experiment, the specific conversion rate of Cu-Au/TiO_2_ prepared using copper nitrate precursor (CuN-Au/TiO_2_) in the batch system was 14 µmol/m^2^·h, whereas, in the microflow system, it reached 268 µmol/m^2^ h.

The specific surface areas determined from N_2_ physisorption for the Cu-Au/TiO_2_ and Fe-Au/TiO_2_ varied from 251 to 547 g/m^2^, which was predominantly controlled by the type of precursor used for synthesis (Table 1). The use of nitrate precursor during the synthesis procedure caused a significant increase (about two-fold) in the specific surface area with reduced crystallinity (Figure 2a) compared to pure TiO_2_. Au phase is not a dominant phase for these catalysts; therefore a small peak of Au is observed. Amorphous phases dominate in CuN-Au/TiO_2_ and FeN-Au/TiO_2_ samples, such phases do not give peaks in the XRD pattern, and on the background of such phases, even a small peak from nanocrystalline gold is easily detectable. The crystallite size of Au for each catalyst are shown in Table 1.

A combination of type II and IV adsorption isotherms was noticed for sol–gel-synthesized TiO_2_ and the bimetallic catalysts with nitrate precursors (Figure 2b) [24]. The H2-type hysteresis loop of these catalysts suggested a complex pore structure [25]. On the other hand, the bimetallic catalyst with acetate precursor (CuA-Au/TiO_2_) showed a type-IV adsorption isotherm [24], suggesting the presence of mesopores. The shape of the hysteresis loops was of type H3, which indicates the presence of aggregates of slit-shaped conical pores composed of primary particles, which can give rise to piled-up pores [26,27,28].

The Tauc plots (Appendix A) revealed that the bandgaps for all the synthesized catalysts were ~3.4 eV. In addition, the BJH (Barrett–Joyner–Halenda) method was used to calculate the average pore size distribution, and the values are provided in Table 1.

The average pore size was lower for samples synthesized using nitrate precursors than for samples that were prepared with acetate precursors. This observation can be explained by a partial modification of the TiO_2_ surface by nitrate decomposition and evolving gaseous NO_x_ during the synthesis procedure. The photocatalytic activity in the microflow system (Figure 1b) of CuN-Au/TiO_2_ and FeN-Au/TiO_2_ accompanied by their larger specific area, as a consequence, may provide more active sites and shorten the bulk diffusion length of charge carriers, thus suppressing bulk recombination [29]. However, the pore volume was much higher for the sample CuN-Au/TiO_2_ (0.4 cm^3^/g) than for FeN-Au/TiO_2_ (0.2 cm^3^/g), even though nitrogen salts were used in both cases during the preparation. This dependency can be caused by steric hindrance during the growth of the TiO_2_ crystallites via Ostwald ripening [30]. The molecular structure of iron (III) nitrate nonahydrate is bigger than copper (II) nitrate trihydrate (computed by PubChem), which consequently causes more steric hindrance.

The synthesized bimetallic catalysts were investigated using transmission electron microscopy (TEM). The composition of the particles was studied using STEM (high-angle annual dark field, HAADF) and EDXS mapping. In the CuA-Au/TiO_2_ sample, the TEM analysis revealed that primary nanoparticles were clustered to form more oversized agglomerates during the formation stage itself (Figure 3). The results of the TEM analysis showed that Ti, O, and Cu are evenly distributed throughout the sample surface (Figure 3d,e).

The application of the HAADF method confirmed the presence of the expected elements, such as Ti, Cu, and Au (Figure 3c–e), and also, from XPS analysis of CuA-Au/TiO_2_, we were able to confirm the presence of 0.1 at% Cu and 0.03 at% Au in the catalyst surface (Appendix A). Gold and copper nanoparticles were evenly distributed in the tested sample (Figure 3b–d) which is very important in terms of the influence on the photocatalytic activity. Based on TEM analysis (Figure 3), UV–Vis DRS spectra (Appendix A), and XPS analysis (Appendix A), it can be deduced that surface incorporation of Cu and Au atoms on the TiO_2_ has occurred. On the UV–Vis spectra, the red shift of absorption for CuA-Au/TiO_2_ in comparison to bare TiO_2_ and no obvious peak from copper nanoparticles (NPs) can be seen. The absence of the peak coming from Cu NPs could be due to the low amount of Cu NPs on the TiO_2_ surface. The same results regarding the slight red shift of absorption to 403 nm for copper nanospheres coupled with TiO_2_ were obtained by Monga et al. and also other research groups [31,32,33]. The distinct peak derived from Au nanoparticles occurred at 560 nm due to the charge transfer from the metal ion to TiO_2_ [32]. The abovementioned absorption in the range of visible irradiation may be due to the localized surface plasmon resonance (LSPR). It has been shown that in the mechanism using LSPR, metal can act as an electron trap and thus inhibit the recombination process of electron-hole pairs [34,35]. The characteristic LSPR band maximum at 555 nm [36] indicates the presence of Au nanoparticles in the CuN-Au/TiO_2_ sample, shown on the UV–Vis DRS spectra (Appendix A).

It can be clearly seen in the TEM pictures (Figure 3b) that the obtained gold nanoparticles have a spherical shape, and the particle size distribution of Au was in the range of 10–40 nm, with the highest contribution of the fraction from 15 to 25 nm and mean size equal to 22.1 nm (Figure 3a). The nanometric size and shape of the obtained Au particles are good enough to provoke LSPR [31,37]. This proves that the particles in nanoscale with a favorable spherical shape were successfully obtained during the synthesis procedure. Gołąbiewska et al. [38] have shown that the spherical shape of gold nanoparticles was very beneficial for increasing the photocatalytic activity in the range of visible irradiation. They compared the photoactivity of different shapes of Au particles deposited on TiO_2_ microspheres and proved that visible light activity decreased in the following order: spheres > rods > stars [38]. It can also be added that in the case of the CuA-Au/TiO_2_ sample, the inter-planar spacing for gold was 0.24 nm, which corresponds to the separation between the (111) lattice planes of Au (Figure 3b), which was consistent with the results obtained by XRD analysis (Table 1).

Very similar to the CuA-Au/TiO_2_ TEM analysis results were the results obtained for the CuN-Au/TiO_2_ sample, where the gold particles have a nanometric size and spherical shape; in addition, they were evenly distributed over the entire sample (Figure 4a–e). The difference in the photocatalytic activity of these samples was probably due to the precursors used (nitrate and acetate). The average size of the Au particles ranges between 10–50 nm, also less than 5% in the range 60–70 nm, with a significant predominance of particles with sizes from 10 to 30 nm, where the mean size was 24.7 nm (Figure 4a) Moreover, the Au particle has planes (111) and (200), which correspond to 0.24 and 0.20 nm, respectively. (Figure 4b).

The TEM micrographs of FeN-Au/TiO_2_ shown in Figure 5 gave a bright observation of the sample in which the particles are much bigger than those obtained in the case of the samples containing copper (the mean sizes of the particles were 22.1 nm for CuN-Au/TiO_2_ and 24.7 nm for CuA-Au/TiO_2_). The particle size was found in the range of up to 60 nm and also in the range from 80 to140 nm, with the mean size equal to 60.6 nm. (Figure 5a). It should be emphasized that in the UV–Vis DRS spectrum (Appendix A), no characteristic peak from the LSPR of gold particles was detected. Similar results were obtained by Duan et al., where despite the presence of gold particles in the sample, confirmed by other analytical methods, the characteristic peak of approx. 550 nm was not observed [39]. Similar results were obtained for the FeN-Au/TiO_2_ sample; there were big agglomerates which caused an uneven distribution of Au particles on the TiO_2_ (Figure 5d,e), which could be the reason for the absence of the peak from Au NPs in the UV–Vis DRS spectra. However, a significant shift of absorption towards the visible spectrum (>400nm) for FeN-Au/TiO_2_ in comparison with unmodified TiO_2_ was observed on UV–Vis DRS (Appendix A), and the presence of Fe 2p 3/2 on the surface of the catalyst (confirmed by XPS analysis) also indicates the probable surface incorporation of Fe and Au atoms on TiO_2_. The aim of introducing the Fe atoms to the sample was to increase the absorption toward the visible light wavelength and thus increase the photocatalytic activity in the visible range [40]. The distribution of Fe atoms observed on TEM connected with UV–Vis DRS and XPS results can testify to the surface incorporation of Fe atoms [41,42].

In the SEM images of the bimetallic TiO_2_ particles, presented in Figure 6, it was observed that the sample prepared using acetate precursor has a more sponge-like surface, with more cavities in the structure, compared to that obtained with nitrate precursor. CuN-Au/TiO_2_ and FeN-Au/TiO_2_ particles were poorly formed with highly irregular shapes.

The use of sonication improves catalyst deposition in a microreactor aided by enhanced mass deposition, which subsequently improves the overall photocatalytic conversion [43]. We link this activity to the higher availability of the active sites upon the deposition on the wall of the microreactor. On the deposition of these catalysts in the microflow system under the influence of ultrasound, the breakage of the agglomerations into smaller sizes was observed. The big aggregates (300–500 nm) in the case of CuA-Au/TiO_2_ disturb the flow of the solution inside the microtube, reducing the activity of the catalyst (Figure 1). The small agglomerations (80–250 nm) and good dispersion in the case of CuN-Au/TiO_2_ and FeN-Au/TiO_2_ (Figure 6) can lead to the improved transfer of light and better interaction of the reagent with the surface of the catalyst. The thickness of the deposited catalyst inside the wall of the microreactor was 5–7 μm, measured with an optical microscope image, hence the better photocatalytic activity in the microflow system for this catalyst.

The synthesized CuA-Au/TiO_2_ catalyst was considered for the oxidation of aromatic alcohols (vanillyl alcohol (VanOH), coniferyl alcohol (ConOH), and cinnamyl alcohol (CinOH) under UV (375 nm) light, as this catalyst showed better activity under UV light for BnOH oxidation (Figure 1a) in the batch system. Before light experiments, dark adsorption studies were performed to determine the adsorption/reactivity, and the duration to reach adsorption equilibrium was 30 min. Strong adsorption of VanOH (37%) and ConOH (30%) was observed on the surface of bimetallic CuA-Au/TiO_2_ in the dark, which might be because of the complex formation between these alcohols with TiO_2_.

After 2 h of the light experiment, BnOH and VanOH conversion were 54% and 95%, respectively (Figure 7b). Comparing the structure of the molecules of BnOH and VanOH (Figure 7a), it can be seen that the main difference between these two alcohols was that VanOH has in its structure an OH group directly connected with the aromatic ring and also a methoxy (OCH_3_) group connected with the next carbon atom of the aromatic ring. In contrast, benzyl alcohol has only an OH group at the end of the alkyl chain, which was not directly connected with the aromatic ring, and no methoxy group. As mentioned above, the OH and OCH_3_ groups in the vanillyl alcohol are electron-donating groups which were providing the electrons to the aromatic ring. An excess of electrons causes lower production of aldehyde, which results in low selectivity towards aldehyde in the case of VanOH (25%) in comparison to BnOH (100%) under UV irradiation (Figure 7b) after 2 h of the light experiment. As the selectivity of producing vanillyl aldehyde (VanAld) was strongly reduced, other products of vanillyl alcohol oxidation were achieved (Appendix A). Furthermore, the lower selectivity can also be ascribed to the strong adsorption (37%) of alcohol on the surface of the catalyst, which can lead to partial oxidation, dimerization, or C-O, C-C coupling products.

Considering ConOH and CinOH alcohols, it can be seen that very high conversions were achieved (95% and 81%, respectively). In the case of ConOH, the selectivity of producing coniferyl aldehyde (ConAld) was very low (5%), with 7% selectivity towards VanAld (Figure 7), with additional products verified by GC-MS analysis (Appendix A). Along with the OCH_3_ and OH groups, ConOH has a double bond in its structure, which can potentially also provide the electrons to the aromatic ring, which causes further reduction, lowering the selectivity. The side products formed are benzaldehyde from CinOH and vanillyl aldehyde from ConOH (partial oxidation) [44], as well as 3-phenoxy benzaldehyde from ConOH (C-O coupling product), which probably was the result of many complex chemical reactions leading to the dimerization of ConOH under the influence of the oxidizing environment (UV radiation) [45]. The conversion of aromatic alcohols and the selectivity to its aldehyde were the main criteria to estimate catalyst performance as the other products were in traces. For CinOH oxidation, the selectivity towards cinnamyl aldehyde (CinAld) was 45%. Furthermore, the catalyst showed highly selective toward BnOH oxidation to BnAld.

After conducting the selective photocatalytic oxidation experiments in UV radiation, the CuA-Au/TiO_2_ photocatalyst was selected for experiments also under visible light in the batch system (Figure 8). Under visible irradiation (515 nm), the satisfactory conversion of the alcohols was obtained only for ConOH and VanOH (93% and 52%, respectively), with 8% and 17% selectivity towards their respective aldehydes, with the formation of other side products. Increased photocatalytic conversion in the range of visible radiation for ConOH and VanOH, with a complete lack of activity for BnOH and CinOH, was probably related to the structures of these alcohols. In the case of BnOH and CinOH, there are no groups that could be a direct source of additional electrons for the aromatic ring and no group which can form complexes with TiO_2_. In order to explain this, UV-–Vis DRS and FTIR analyses were performed for the aromatic alcohol-adsorbed TiO_2_ complexes (Figure 9 and Figure 10) [46,47].

As a result of the adsorption of ConOH and VanOH on the TiO_2_ catalyst, a significant change in the color of the catalyst from white to yellow can clearly be seen, which resulted in a shift of absorption toward visible radiation (Figure 9). A plausible reason can be complex formation by the ligand-to-metal charge transfer (LMCT) [48,49]. During this process, electrons are transferred from the highest occupied molecular orbital (HOMO) of substrates/adsorbates to the conduction band of TiO_2_ upon visible light irradiation. (Figure 1) [46]. Consequently, the samples forming the colorful complexes exhibited an excellent conversion of aromatic alcohols under visible light (Figure 10).

The formation of complexes between TiO_2_ and ConOH and TiO_2_ and VanOH was also supported by the FTIR analysis results (Figure 10). Characteristic IR bands for alcohols have also been observed for alcohol-adsorbed titania samples, especially in TiO_2_-VanOH and TiO_2_-ConOH (Appendix A). This provided an indication of complex formation via the adsorption of alcohol.

It was mentioned above that the adsorption of alcohols (VanOH and ConOH) on the surface of TiO_2_ occurred by a dissociative mechanism, which means that OH and O-CH_3_ groups interacted between titania and aromatic alcohols. In that case, alcohol and TiO_2_ would be linked via the C-O bond. The presence of C-O stretching vibrations was observed at 1000–1200 cm^−1^ (Appendix A), which can be seen in the case of TiO_2_-VanOH and TiO_2_-ConOH; two bands were observed in this region (1118 and 1158 cm^−1^). The bands in the range of 1000–1050 cm^−1^ were assigned to the presence of =C-H bonds derived from the aromatic ring (Figure 10 and
Appendix A). The bands around 1460 cm^−1^ originated from CH_2_ vibrations and around 1640 cm^−1^ from C=C [46]. Similar spectra were observed for bimetallic CuA-Au/TiO_2_ as well, confirming the formation of the complex with the above aromatic alcohol (Appendix A), which confirms that the metals did not take part in this complex formation. It is worth adding that the oxidation of VanOH and ConOH was successful under visible irradiation due to the formation of the abovementioned complex with TiO_2_. To further confirm the role of the OH group in LMCT complex formation, the synthesized TiO_2_ was calcined at a high temperature (600 °C) to remove surface OH through condensation (Appendix A). Interestingly, this catalyst was found inactive, suggesting the OH groups are crucial for LMCT complex formation and the visible light activity of the catalyst.

The experiment with the CuN-Au/TiO_2_-coated microreactor for other aromatic alcohols oxidation was performed under visible light irradiation as (1) CuN-Au/TiO_2_ in the microflow system showed better activity (Figure 1) and (2) the other aromatic alcohols showed better conversion under visible light (Figure 8). Like the batch experiment, BnOH and CinOH did not show any activity in the microflow system under visible light, whereas ConOH and VanOH did. The specific conversion rate of ConOH was higher (198 μmol/m^2^ h) compared to VanOH (46 μmol/m^2^ h), with 11 % and 10 % selectivity towards their respective aldehydes (Figure 11). Other side products, such as batch experiments, were also observed.

After deposition, the catalyst’s surface was more selective towards ConAld for ConOH oxidation than in the batch system (8%). The elevated selectivity in the batch system compared to the microflow system might be because of the higher availability of the catalyst’s active sites upon deposition. Additional side products (as discussed above) were confirmed from GC/MS; however, because of the trace amount of some products, it was not easy to calculate the selectivity towards them (Appendix A). The sol–gel-synthesized CuN-Au/TiO_2_ catalyst showed promising activity in the batch system as well as in the microflow system for ConOH oxidation. Furthermore, from the catalyst-deposited microtubes, no leaching was confirmed with the ED-XRF analysis after 60 min of the experiment (Appendix A). The recycling test with the best performing catalyst (CuA-Au/TiO_2_) and the one which showed lower activity (TiO_2_) in the batch system were performed by washing the catalyst with acetonitrile and water, and the activity of the catalyst was retained (Appendix A).

## 3. Materials and Methods

### 3.1. Materials

Titanium (IV) tetraisopropoxide (TTIP, 98%, ACROS ORGANICS, Geel, Belgium), benzyl alcohol (BnOH, 99.5%, CHEMPUR, Piekary Śląskie, Poland), coniferyl alcohol (ConOH, 98%, ABCR, Karlsruhe, Germany), cinnamyl alcohol (CinOH, 98%, ACROS ORGANICS, NJ, USA), vanillyl alcohol (VanOH, 98%, ABCR, Karlsruhe, Germany), ethanol (EtOH, 99.8%, POCH, Gliwice, Poland), acetonitrile (ACN, HPLC grade, POCH, Gliwice, Poland), and propan-2-ol (99.7%, CHEMPUR, Piekary Śląskie, Poland) were used as received. Copper (II) acetate monohydrate (Cu(CH_3_COO)_2_·H_2_O, 98%, ABCR, Karlsruhe, Germany), iron (III) nitrate nonahydrate (Fe(NO_3_)_3_·9H_2_O, 98%, ABCR, Karlsruhe, Germany), copper (II) nitrate trihydrate (Cu(NO_3_)_2_·3H_2_O, pure, CHEMPUR, Piekary Śląskie, Poland), and tetrachloroauric (III) acid trihydrate (HAuCl_4_·3H_2_O, >99.5%, ROTH, Karlsruhe, Germany) were used as received as metal precursors.

### 3.2. Catalyst Synthesis

The bimetallic Cu-Au/TiO_2_ and Fe-Au/TiO_2_ nanoparticles were prepared based on the previously established sol–gel method [50]. The required amount of metal precursors (ratio of atomic percent of Au to Cu or Fe was adjusted to 1:4) were added to 15 mL of isopropanol and stirred (magnetic stirrer) at 600 rpm. Under the rotational conditions, 4 mL of TTIP was added dropwise to the mixture. Subsequently, milli-Q water (5 mL) was added to the solution at a rate of 0.167 mL/min using a syringe infusion pump (Programmable Double Syringe Pump (WPI), NE-4000). After 6 h of aging with mixing at 1000 rpm, the resulting solid samples were obtained by centrifugation, washing with deionized water (2 times) and ethanol (1 time), and drying at 80 °C in an oven for 24 h.

### 3.3. Microreactor Preparation

A visible light transparent perfluoroalkoxy alkane (PFA, 0.8 mm ID, BOLA: S 1811-02) tube was used as a microreactor [50,51]. A 0.5 g/L concentration (previously optimized [50]) of nanoparticles was dispersed in Milli-Q water by ultrasonication for 15 min using an ultrasonic bath (Elma Elmasonic P, 37 kHz, 70% power). Twenty mL of the homogeneous nanoparticle suspension was passed through the cleaned PFA microtube under the influence of ultrasound using a syringe pump. The spiralized fragment (Figure 12b) was the effective length subjected to the ultrasound treatment (Elma Elmasonic P, 37 kHz, 100% power) for 75 min (the flow rate of the suspension was 0.26 mL/min). The tube was placed in the oven for 24 h at 80 °C, and later cleaned by passing Milli-Q water and ethanol, dried with airflow, and then placed in the oven for 1 h at 80 °C. After this procedure, the catalyst-coated PFA tube was used for photocatalytic experiments (Figure 12b).

### 3.4. Catalytic Performance Test

The photocatalytic oxidation of the aromatic alcohols over the synthesized catalysts was carried out in batch (Figure 12a) and microflow reactors (Figure 12b, Appendix A). UV-LED and Vis-LED systems were used as light sources (375 nm and 515 nm wavelength, respectively). The flow rate was set at 0.167 mL/min (after optimization) to reproduce enough space–time according to the reactor’s dimensions, and the whole experiment was carried out for 60 min [50]. Experiments in the batch photocatalytic reactor were performed using 20 mL of the reaction solution and 0.5 g/L of catalyst concentration for 60 min under UV light. Alcohol conversion, the specific conversion rate, and selectivity to each product were calculated according to the following equations:Conversion (%) = ((Converted moles of aromatic alcohols)/(Initial moles of aromatic alcohols)) × 100% 
Selectivity (%) = ((Produced moles of aromatic aldehydes)/(Converted moles of aromatic alcohols)) × 100%
Specific Conversion Rate (µmol/m^2^·h) = (C_Oho_ − C_OHt_ )/(S_C_ × time)
where C_OHo_ = initial concentration of aromatic alcohols, C_OHt_ = concentration of aromatic alcohols after time, t, and S_C_ is the specific surface area (surface area multiplied by the catalyst concentration) of the catalyst taking part in the photocatalytic conversion.

### 3.5. Characterization

The synthesized samples were characterized using diffuse reflectance spectroscopy (DRS, Shimadzu UV-2600i), and the bandgap was calculated from the Tauc plot. Powder X-ray diffraction (XRD) measurements were performed employing the Bragg–Brentano configuration. This type of arrangement was provided using a PANalytical Empyrean diffraction platform, powered at 40 kV × 40 mA and equipped with a vertical goniometer, with theta–theta geometry using Ni-filtered CuKα (λ = 1.5418 Å) radiation. The elemental maps of the samples were also obtained by FEI Nova Nanolab 200 scanning electron microscopy (SEM). The textural properties of TiO_2_ were determined by N_2_ physisorption using a micromeritics automated system (Micromeritics Instrument Corporation, Norcross, GA, USA) with the Brunauer–Emmet–Teller (BET) and the Barret–Joyner–Halenda (BJH) methods. Before adsorption, the samples were degassed under vacuum (0.1 Pa) for 12 h at 80 °C. The presence of functional groups on the surface of the catalyst and substrate was determined using a Bruker Tensor II Fourier transform (FT) IR spectrometer.

The samples collected from the outlet of the catalyst-deposited PFA capillary were examined using the energy dispersive X-ray fluorescence (EDXRF) spectrometer (Mini- Pal 4, PANalytical Co., Malvern, UK) with a Rh tube and silicon drift detector to check the titania residual and other metals. The spectra were collected in an air atmosphere, without a filter, at a tube voltage of 30 kV. To identify and quantify the alcohols, aldehydes, and acids present, the collected samples were analyzed using high-pressure liquid chromatography (HPLC, Waters) with mass spectrometry using a mobile phase containing a mixture of organic solvents and a 0.05% H_3_PO_4_ (5M) aqueous solution (CH_3_CN:CH_3_OH: H_2_O = 20:2.5:77.5 *v*/*v*).

## 4. Conclusions

In this study, we have made an attempt to promote green chemistry by improving the utilization of lignin—an important waste from the paper and pulp industry, which otherwise would have been disposed of, thus contaminating the environment. To this end, we propose to improve the conversion of the model alcohols by targeting and improving the catalytic activity of TiO_2_ doped with bimetals such as Cu-Au, and Fe-Au. In the batch system, as high as 100% benzaldehyde (BnAld) selectivity was obtained for benzyl alcohol (BnOH) conversion of ~60% using CuA-Au/TiO_2_ photocatalyst. The high conversion could be explained by the high average pore size (6.8 nm) and better crystallinity of the CuA-Au/TiO_2_ catalyst, which were confirmed in subsequent XRD and N_2_ physisorption analysis.

Our studies further revealed that the LMCT complex formation of TiO_2_ with the methoxy (OCH_3_) and OH groups (directly connected with the aromatic ring) exists in the structure of coniferyl alcohol (ConOH) and vanillyl alcohol (VanOH), which was crucial to activate the catalyst under visible light. This hypothesis has been further corroborated by UV–Vis and FTIR studies. Additionally, the whole process was green/environment-friendly as the catalyst synthesis process did not include any high-temperature calcination steps, unlike commercial P25 TiO_2_, and the photocatalytic selective oxidation route was additive-free (no additional molecular oxygen). We also developed a catalyst-decorated microreactor using ultrasonic irradiation, which helped to increase the turbulence of the liquid phase and to improve the active surface area of our catalyst via the de-agglomeration and fragmentation of the nanoparticles.

In a broader context, we believe that the presented work demonstrates the potential of an ultrasonic-assisted bimetallic TiO_2_ wall-coated microreactor for selective oxidation of lignin-based model compounds using solar energy and this will serve as a conceptual blueprint for further developments.

## Data Availability

Not applicable.

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
