# Peer review of "Bimetallic TiO2 Nanoparticles for Lignin-Based Model Compounds Valorization by Integrating an Optocatalytic Flow-Microreactor"

_molecules, 2022, doi:10.3390/molecules27248731_

Round 1
Author Response
We thank the referees for their time and in-depth insight into our manuscript, which have helped us to improve the quality of our work. All their comments are addressed, and suggestions have been incorporated into the revised manuscript. The changes made in the manuscript are highlighted in yellow (with ‘Track Changes’ on), and a point-by-point response is appended below for your convenient reading.

Reviewer 2 Report
Few comments and changes are proposed at the owrd document.

Author Response
We thank the referees for their time and in-depth insight into our manuscript, which have helped us to improve the quality of our work. All their comments are addressed and suggestions have been incorporated into the revised manuscript. The changes made in the manuscript are highlighted in yellow (with ‘Track Changes’ on) and a point-by-point response is appended below for your convenient reading.

Reviewer 3 Report
S.R. Pradhan et al presented the following manuscript entitled as ‘’ Bimetallic TiO2 Nanoparticles for Lignin-Based Model Com[1]3 pounds Valorization by Integrating Optocatalytic Flow-Micro-4 reactor.
Authors synthesize bimetallic TiO2 nanomaterial using Au with Cu and Fe with the ratio (1:4) using Cupper acetate, cupper nitrate and iron nitrate precursor. New scientific approach to use PFA and its inner surface modification with CuA-Au/TiO2, CuN-Au/TiO2 and FeN-Au/TiO2 nanomaterial. Obtained data are promising. This manuscript needs to explain the following queries before consideration.
Introduction:
1- Line 29: Sentence must not start with ‘Because’. Need to revise the sentence and start with proper background of TiO2 material.
2- Line 46-48, ‘As is well known, bimetallic catalysts show tunable and synergistic effects for conventional heterogeneous catalysis compared to their monometallic counterparts.’ It is in need to revise.
3- Need to include some crude data finding in Abstract and introduction.
Material and Method:
1- In material, please add the city and county for each chemical.
2- In catalytic synthesis section: Why did the authors choose Au: Cu and Fe (1:4) and isopropanol?
3- In in microreactor preparation section: Why was PFA 0.8 mm choose in term benefit for this study?
4- PFA pipe usually somehow opaque, as bimetallic nanoparticles were coated inside the PFA tube it will enhance. How did UV-Vis light pass through the tube (as being opaque) and bi-metallic TiO2 work as photocatalyst?
5- What are the possibilities for the conversion CuO or Fe2O3 or Fe3O4 nanostructure with Au/TiO2 nanoparticles as samples were heated at 80 0C ?
Results and discussion:
1- In Figure 2(a), CuA-Au/TiO2 nanomaterials shown high conversion while CuN-Au/TiO2 lower. However, initially conversion % was same.
2- XRD data is not interpreted and Fig. 3(a) is not mentioned in the text?
3- Need to include the recyclability of each system.
4- One comparative table with literature should include in the manuscript.
Conclusion:
Conclusion must revise and improvise with core finding data.
Author Response

(The authors gave the same response as above.)

Round 2
Reviewer 1 Report
The authors made the requested changes and reply to all questions. I recommend it for publication in the present form